# Activation of Cdc42 GTPase upon CRY2-Induced Cortical Recruitment Is Antagonized by GAPs in Fission Yeast

**DOI:** 10.3390/cells9092089

**Published:** 2020-09-12

**Authors:** Iker Lamas, Nathalie Weber, Sophie G. Martin

**Affiliations:** Department of Fundamental Microbiology, Faculty of Biology and Medicine, University of Lausanne, Biophore building, 1015 Lausanne, Switzerland; iker.lamasherrera@unil.ch (I.L.); nadliweber@hotmail.com (N.W.)

**Keywords:** Cdc42, GTPase activating protein (GAP), cell polarity, fission yeast *Schizosaccharomyces pombe*, CRY2-CIBN, optogenetics, clustering, positive feedback, pattern formation

## Abstract

The small GTPase Cdc42 is critical for cell polarization in eukaryotic cells. In rod-shaped fission yeast *Schizosaccharomyces pombe* cells, active GTP-bound Cdc42 promotes polarized growth at cell poles, while inactive Cdc42-GDP localizes ubiquitously also along cell sides. Zones of Cdc42 activity are maintained by positive feedback amplification involving the formation of a complex between Cdc42-GTP, the scaffold Scd2, and the guanine nucleotide exchange factor (GEF) Scd1, which promotes the activation of more Cdc42. Here, we use the CRY2-CIB1 optogenetic system to recruit and cluster a cytosolic Cdc42 variant at the plasma membrane and show that this leads to its moderate activation also on cell sides. Surprisingly, Scd2, which binds Cdc42-GTP, is still recruited to CRY2-Cdc42 clusters at cell sides in individual deletion of the GEFs Scd1 or Gef1. We show that activated Cdc42 clusters at cell sides are able to recruit Scd1, dependent on the scaffold Scd2. However, Cdc42 activity is not amplified by positive feedback and does not lead to morphogenetic changes, due to antagonistic activity of the GTPase activating protein Rga4. Thus, the cell architecture is robust to moderate activation of Cdc42 at cell sides.

## 1. Introduction

In eukaryotes, the small Rho-family GTPase Cdc42 is a highly conserved regulator of cell morphogenesis, proliferation, and differentiation. Prenylation of Cdc42′s C-terminal CAAX motif underlies its association with the plasma membrane, where it functions as a molecular switch that alternates between GTP-bound, active and GDP-bound, inactive states. Activation of Rho GTPases relies on the activity of guanine nucleotide exchange factors (GEFs), while their intrinsic GTPase activity is enhanced by GTPase activating proteins (GAPs) to return them to the inactive state. GDP-bound Cdc42 also binds guanine-nucleotide dissociation inhibitors (GDI), which both block the exchange of GDP by GTP and solubilize Cdc42-GDP in the cytosol [1,2,3].

In the fission yeast *Schizosaccharomyces pombe,* Cdc42 is active at sites of polarized growth during vegetative and sexual life cycles. GTP-loading is promoted by two GEFs, Scd1 and Gef1. Scd1, which localizes to cell poles, receives information from the upstream Ras1 GTPase signal and mediates feedback control through the scaffolding activity of Scd2 [4,5,6]. For this, Scd1 forms a quaternary complex with Cdc42-GTP, the Pak1 kinase effector and Scd2 [7,8], which leads in vivo to the positive feedback activation of other Cdc42 molecules, as shown in our recent work using optogenetic strategies [6]. The second GEF, Gef1, which localizes to cell poles only in some conditions, promotes Cdc42 activation in response to stress and becomes essential only in absence of Scd1 [9,10,11,12]. Three GAPs, namely Rga4, Rga6, and Rga3, enhance the intrinsic GTP hydrolytic activity of Cdc42 [13,14,15]. Rga4 and Rga6 GAPs localize at cell sides, where growth does not occur in non-stressed cells, whereas Rga3 localizes at sites of active growth (cell poles). Fission yeast cells also express a GDI, called Rdi1, though Cdc42 localization and dynamics are not strongly perturbed in its absence [4,16].

Recently, optogenetic studies revealed a novel mechanism that triggers the activation of small GTPases in mammalian cells: Human Rac1 and RhoA, which belong to the same Rho GTPase family as Cdc42, were shown to become active at the cell cortex upon light-dependent cytosolic clustering [17]. In these experiments, the small GTPases were fused to CRY2PHR, the photolyase homology region of *A. thaliana* cryptochrome 2, which oligomerizes upon blue light exposure. Artificially clustered RhoA induced RhoA signalling-dependent cytoskeletal re-organization and membrane retraction in human cells, suggesting that oligomerization promotes RhoA activation [17]. Ras and Ras-like GTPases are well known to form nanoclusters and dimers at the membrane to activate signal transduction [18,19,20]. Several Rho-family GTPases, including RhoA, Rac1, Rac2 and Cdc42, were also shown to form dimers or oligomers through homophilic interactions of their polybasic region adjacent to the C-terminal CAAX motif [21,22]. While oligomerization of GTP-bound Cdc42 and Rac1 increases their GTPase activity in vitro, the physiological relevance of clustering of these small GTPases remains to be investigated [22]. In vivo, Rac1-GTP oligomers have been shown to contain several dozen Rac1 molecules together with charged phospholipids and appear to promote signal transduction [21,22,23,24]. Cdc42 also forms nanoclusters in *Saccharomyces cerevisiae* cells [25,26]. These nanoclusters show an anisotropic distribution: they accumulate and exhibit larger sizes at cortical sites of polarized growth, in a manner dependent on the scaffold protein Bem1 and anionic membrane lipids [25,27]. Because Bem1 also acts as scaffold that bridges Cdc42-GTP to its GEF and promotes positive feedback activation of Cdc42, Cdc42 nanoclusters may promote Cdc42 feedback activation, though this has not been tested.

In this work, we used an artificial optogenetic strategy to induce the recruitment and clustering of Cdc42 at the plasma membrane of fission yeast cells. We built on our recent work that used the CRY2-CIB1 optogenetic system to probe the positive feedback of Cdc42 [6]. The CRY2-CIB1 system relies on the blue light-induced binding of CRY2PHR (simply denoted CRY2 below) to the N-terminal part of CRY2-binding partner CIB1 (CIBN) [28]. Blue light also induces the formation of CRY2 oligomers [17]. We fused CRY2 to a cytosolic variant of Cdc42 (Cdc42^∆CaaX^) and co-expressed CIBN linked to the membrane-associated RitC anchor. In our earlier study, we showed that cortical recruitment of a GTP-locked, constitutively active Cdc42 variant (CRY2-Cdc42^Q61L,∆CaaX^) led to the Scd2-dependent co-recruitment of its GEF Scd1 and accumulation of endogenous Cdc42, demonstrating feedback amplification [6]. Surprisingly, we also found that cortical recruitment of CRY2-Cdc42^∆CaaX^ (not GTP-locked) also induced the co-recruitment of Scd2, suggesting the activation of CRY2-Cdc42^∆CaaX^. In this work, we confirm that CRY2-dependent recruitment of Cdc42^∆CaaX^ at lateral sites, where Cdc42 is normally inactive, promotes its activation. We show that activated clustered Cdc42 is able to recruit its GEF Scd1 through the scaffold Scd2, suggesting that positive feedback is initiated. However, the activation is efficiently countered by Rga4 GAP-mediated Cdc42 inactivation, and does not lead to cell shape alteration, showing the robustness of the cell polarization system.

## 2. Results and Discussion

### 2.1. Weak Activation of CRY2-Cdc42 at the Cell Cortex

To better characterize CRY2-Cdc42^∆CaaX^, we first measured its kinetics of recruitment to CIBN-RitC at the plasma membrane. Similar to rates measured for CRY2, CRY2-Cdc42^∆CaaX^ showed a half-time of protein recruitment to the cortex < 1 s and independent of the length of the blue light (488 nm) pulses (30 GFP pulses of 50 ms = 0.92 s ± 0.24 s; 22 GFP pulses of 250 ms = 0.98 s ± 0.25 s; 17 GFP pulses of 500 ms = 0.99 s ± 0.31 s; Appendix A). CRY2-Cdc42^∆CaaX^ cells did not exhibit any morphological defects and grew in a bipolar fashion in the dark (Appendix A). In blue-light, CRY2-Cdc42^∆CaaX^ cells maintained their characteristic rod-shape and continued growing from the cell tips (Appendix A, green cells), while cells with GTP-locked CRY2-Cdc42^Q61L,∆CaaX^ rounded up indicating isotropic growth (Appendix A, blue cells; [6]). These evidences initially suggested that the recruitment of CRY2-Cdc42^∆CaaX^ to the cell cortex was innocuous and unable to bias the endogenous Cdc42 and its regulatory network.

We monitored the distribution of Cdc42-GTP using three GFP-tagged markers that specifically associate with Cdc42-GTP: the scaffold Scd2 [7,29], the CRIB bioreporter (Cdc42-Rac1-interactive-binding domain, [30]), and the Cdc42 effector Pak1, which also contains a CRIB domain. These proteins and probe are normally only detected at the poles and division sites of yeast cells, as well as weakly in the nucleus for the first two. As previously described [6], upon blue light-dependent recruitment of CRY2-Cdc42^∆CaaX^ to the plasma membrane, Scd2-GFP formed stable foci at the cell sides, which increased progressively in intensity and coincided with CRY2-Cdc42^∆CaaX^ clusters, while Scd2-GFP intensity decreased at cell poles (Figure 1A–D and Appendix A). We had previously shown that Scd2 was strongly recruited by GTP-locked CRY2- Cdc42^Q61L,∆CaaX^ but not GDP-locked CRY2- Cdc42^T17N,∆CaaX^ [6]. Indeed, we confirmed that CRY2-Cdc42^∆CaaX-T17N^ does not lead to Scd2 foci at cell sides, indicating that CRY2- Cdc42^∆CaaX^ must be in the GTP-bound form to recruit Scd2 (Figure 1A). CRIB-3GFP also formed dim foci at the cell sides, which became visible 40–60s after light stimulation (Figure 1A,E and Appendix A). The CRIB-3GFP side signal was however weaker and delayed relative to that observed upon light-induced recruitment of GTP-locked CRY2-Cdc42^Q61L,∆CaaX^ (Figure 1E; [6]). Indeed, recruitment of CRIB-3GFP was only statistically significant in the second half of the 87 s-time-lapse (see materials and methods). This side recruitment was also mirrored by a reduction of the CRIB signal at the cell poles (Figure 1F). The Pak1-sfGFP traces also showed an upward trajectory on cell sides and a downward trajectory at cell poles but were not statistically different from negative control after 87 s (Figure 1A,G,H). We note, however, that the higher levels of both CRIB and Pak1 on cell sides were marginally statistically significant after 30 min illumination (*p* = 0.04; see Figure 5B,C). Based on the recruitment of Scd2 by CRY2-Cdc42^∆CaaX^ on cell sides, these data suggest that the heterologous Cdc42 moiety within the CRY2-Cdc42^∆CaaX^ system is transiently activated when recruited in clusters at the cell sides and sufficient to alter the endogenous sites of Cdc42 activity at cell poles. We hypothesize that the stronger Scd2 than CRIB and Pak1 signals reflect a more stable binding, likely stabilized by additional association, for instance, to anionic lipids [27].

As an alternative strategy to increase Cdc42 levels at the plasma membrane, we overexpressed Cdc42. In this experiment, we used the functional, internally tagged *cdc42-mCherry^SW^* allele [4] expressed under the *p^act1^* promoter in cells lacking the endogenous gene, which allowed us to quantify the global increase in expression levels at 3.3-fold (Appendix A). The Cdc42 level increase was roughly uniform around the cell cortex (not shown). Cdc42 overexpression also led to a 1.2-fold increase in the expression of the CRIB-3GFP reporter (expressed under the *p^pak1^* promoter, Appendix A). Cdc42 overexpression led to a small increase in CRIB signal at cell poles (even after correction by the 1.2-fold increase in probe expression) and a small increase in cell length (*p^act1^-cdc42* cell length = 14.2 ± 1 µm vs. WT cell length = 13.4 ± 1.1 µm, t-test *p*-value = 2.8 × 10^−5^; p^act1^-cdc42 cell width = 3.8 ± 0.3 µm vs. WT cell width = 3.8 ± 0.3 µm, *t*-test *p*-value = 0.96), suggesting increased Cdc42 activity at cell poles. However, Cdc42 overexpression had no effect on CRIB-3GFP levels at cell sides (Appendix A). We conclude that activation of CRY2-Cdc42^∆CaaX^ on cell sides is not simply a consequence of overexpression but may be due to other changes imposed by CRY2 activation. It is possible that CRY2-dependent clustering of Cdc42 directly causes the activation of the GTPase, as has been proposed for other GTPases [17], though unknown mechanism. Alternatively, clustering may have indirect effects, such as slowing down Cdc42 dynamics, which influence its activation cycle.

### 2.2. CRY2-Cdc42 Activation in Absence of Cdc42 GEFs

To probe the mode of CRY2-Cdc42^∆CaaX^ activation, we repeated the optogenetic experiments above in strains lacking the Cdc42 GEF Scd1. In *scd1∆* cells, Pak1-sfGFP was not detected at cell sides, similar as in wildtype cells. We also observed only rare CRIB-3GFP dots, and no significant increase in CRIB levels at the sides nor decrease at cell poles of *scd1∆* cells (Figure 2A–C), suggesting that Scd1 participates in CRY2-Cdc42^∆CaaX^ activation. However, Scd2-GFP was still recruited to cell sides and decreased from cell poles (Figure 2A–C). Because Scd2 recruitment is strictly dependent on the GTP-bound form of Cdc42 (see Figure 1A; [6,7,29]), this suggests that CRY2-Cdc42^∆CaaX^ may still be active in these cells. We thus probed the role of the second Cdc42 GEF Gef1. In *gef1∆* cells, Scd2-GFP accumulation at cell sides and decrease at cell poles exhibited similar dynamics as in WT cells (Figure 2D–F). It is possible that the two GEFs work redundantly in this situation, a hypothesis difficult to test due to the lethality of *scd1∆ gef1∆* double mutants [11,12]. An alternative hypothesis, which we do not favour, is that Cdc42 clustering through CRY2 binding may promote Scd2 recruitment independently of its activation.

### 2.3. CRY2-Cdc42 Promotes Recruitment of Its GEF Scd1 in Scd2 Scaffold-Dependent Manner

Because Cdc42-GTP promotes the recruitment of its GEF Scd1 for feedback amplification of Cdc42 activation [6], we probed whether CRY2-Cdc42^∆CaaX^ induces Scd1 recruitment. Indeed, Scd1 formed weak foci at cell sides upon blue-light activation (Figure 3A,B), similar to the CRIB-3GFP foci observed in CRY2-Cdc42^∆CaaX^ cells (see Figure 1A and Appendix A). The appearance of Scd1 foci at cell sides was also mirrored by a decrease of Scd1-3GFP at the cell tips (Figure 3C). Scd1 recruitment was dependent on the scaffold Scd2, as no cell side accumulation of Scd1-3GFP, nor decrease at cell tips, was detected in *scd2∆* cells (Figure 3D–F). These data suggest that the activated CRY2-Cdc42^∆CaaX^ is poised to trigger the positive feedback leading to recruitment of its GEF Scd1.

Although *scd2* deletion abolished Scd1 recruitment, it did not substantially affect the accumulation of the CRIB probe (Figure 3D,G), indicating that CRY2-Cdc42^∆CaaX^ is still activated in these cells. This observation is in agreement with the finding that CRY2-Cdc42^∆CaaX^ may be activated independently of Scd1. Because CRIB intensity on cell sides was not reduced in *scd2∆* cells (Figure 3H), we conclude that the scaffold-dependent recruitment of the GEF by CRY2-Cdc42^∆CaaX^ does not play a major role in amplifying Cdc42 activation at cell sides.

### 2.4. The Cdc42 GAP Rga4 Prevents Isotropic Growth of CRY2-Cdc42^∆CaaX^ Cells

If CRY2-Cdc42^∆CaaX^ is activated at cell sides and recruits its own GEF, why is Cdc42 activity not further amplified by the positive feedback mechanism and does not lead to cell shape changes? Indeed, even long-term growth of CRY2-Cdc42 cells in the light did not change their cell length and width, or aspect ratio, similar to control CRY2 cells. By contrast, constitutive cortical localization of CRY2-Cdc42^Q61L,∆CaaX^ by growth in light conditions led to a significant increase in cell width and decrease in cell length, yielding a reduced aspect ratio (Figure 4A–C).

We hypothesized that the activation of CRY2-Cdc42^∆CaaX^ at cell sides is rapidly counteracted by negative regulators. We focused our attention on the three Cdc42 GAPs Rga3, Rga4 and Rga6, the GDI protein Rdi1 and the Ras1 GAP Gap1. Rga3, Rga4, and Rga6 directly promote Cdc42-GTP hydrolysis [13,14,15]. Rdi1 may promote Cdc42 extraction from the membrane, although previous work showed that it is largely dispensable for Cdc42 dynamics in *S. pombe* [4,16]. Gap1 directly promotes Ras1-GTP hydrolysis [31]. As Ras1 promotes Scd1 activation and is uniformly active at the plasma membrane in *gap1∆* [6,31,32], we hypothesized Scd1 activation on cell sides may be amplified in this mutant. We constructed single and most double deletion mutants expressing either of CRY2-Cdc42^∆CaaX^, CRY2-Cdc42^Q61L,∆CaaX,^ as positive control or CRY2 alone as negative control. We then measured the cell length and cell width of calcofluor-stained dividing cells after at least 14h of exponential growth in light conditions and calculated aspect ratios (Figure 4A).

To estimate the change in aspect ratio upon Cdc42 lateral recruitment while taking into account the initial shape of the cell, we normalized the aspect ratios from cells recruiting Cdc42 to those expressing only CRY2 (Figure 4B). CRY2-Cdc42^Q61L,∆CaaX^ led to a >2-fold reduction in aspect ratio in WT, *rga6∆*, *rdi1∆* and *gap1∆* cells, but had less effect on cell shape change in single and double *rga4∆* mutants, perhaps in part due to the already wider cell shape of *rga4∆* cells [14]. Interestingly, CRY2-Cdc42^∆CaaX^ had little effect on aspect ratio in WT or any single mutants, except in *rga4∆* cells, which became significantly rounder. Similar, more marked effects were also observed in combinations of *rga4∆* with *rga6∆* or *rdi1∆*. The effect of CRY2-Cdc42^∆CaaX^ on the shape of these mutants can also readily be observed in plots of cell width to cell length, with the CRY2-Cdc42^∆CaaX^ cell population placed at an intermediate position between the negative CRY2 and positive CRY2-Cdc42^Q61L,∆CaaX^ controls (Figure 4C). These data indicate that the optogenetic-dependent Cdc42 activation is counteracted by Cdc42 GAPs placed at cell sides.

We hypothesized that Rga4 and Rga6 directly antagonize Cdc42 activity at cell sides. To probe this idea directly, we imaged CRIB-3GFP and Pak1-sfGFP for 87s immediately after CRY2-Cdc42^∆CaaX^ recruitment. Both of these Cdc42-GTP interactors showed significantly increased levels at the sides of *rga4∆ rga6∆* cells (Figure 5A). The data for Pak1 contrast with what we observed in wildtype cells, where no significant increase was detected at this timepoint (see Figure 1G). To strengthen these data further, we probed for CRIB-3GFP and Pak1-sfGFP localization at cell sides at a later timepoint, after 30 min of continuous white-light illumination, similar to the light conditions used to assess cell shape changes. Compared to the small increase of CRIB-3GFP and Pak1-sfGFP at the sides of wildtype cells in these conditions, *rga4∆ rga6∆* mutants showed higher increase, which was highly statistically significant (Figure 5B,C). These data support the view that Cdc42 GAPs at cell sides antagonize the CRY2-Cdc42^∆CaaX^ activation to prevent cell widening. A second, non-mutually exclusive possibility is that the cell shape change observed in *rga4∆* cells reflects the weakening of polarity at the native sites at cell poles, as shown by the decrease of Scd2 and CRIB at cell poles (see Figure 1D,F). Thus, both local and competition effects with native polarity sites may combine to alter the shape of CRY2-Cdc42^∆CaaX^-expressing *rga4∆* cells.

In summary, the data presented in this work show that CRY2-clustered Cdc42 is ectopically activated at cell sides and leads to an alteration of the native polarity sites at cell poles. It is possible that CRY2-tagging interferes with Cdc42 GTPase activity, such that CRY2-Cdc42^∆CaaX^ represents a slightly activated Cdc42 variant. Alternatively, the CRY2-dependent clustering may lead to Cdc42 activation, as proposed for other GTPases, such as Rac1 and RhoA in mammalian cells [17]. CRY2-clustered Cdc42 leads to very clear Scd2 recruitment, but much weaker CRIB and Pak1 recruitment, raising the question of why these Cdc42-GTP binding proteins behave differently. We hypothesize that this is explained by the stabilization of Scd2 localization through additional interactions for instance with membrane lipids. Interestingly, the activity level of CRY2-Cdc42^∆CaaX^ at cell sides is sufficient to recruit the GEF Scd1, whose recruitment to Cdc42-GTP depends on the scaffold Scd2 [6]. However, positive feedback does not appear to become established as Cdc42 activity levels at cell sides are not altered in absence of Scd2. It is possible that the non-physiological linkage of Cdc42 to the plasma membrane (through RitC-CIBN-CRY2 binding instead of through the normal prenyl group) undermines the positive feedback, although we previously showed that a similarly engineered constitutively active Cdc42 construct did trigger feedback-dependent recruitment of endogenous Cdc42 [6]. We favour the view that cellular regions may not be equally permissive to Cdc42 activity feedback amplification. For instance, the absence of Ras1 activity on cell sides may reduce the effectiveness of the feedback [6]. Our data further indicate that the inefficient positive feedback at cell sides may be due to the action of the cell side-localized Cdc42 GAP Rga4, which we have shown antagonizes the effect of CRY2-Cdc42^∆CaaX^ on effector recruitment and cell morphogenesis. Additional mechanisms preventing growth on cell sides are also being proposed [33]. Thus, one critical question for future research is to better understand the many layers that confer robustness to cell morphogenesis.

## 3. Materials and Methods

### 3.1. Strains, Media, and Growth Conditions

Strains used in this study are listed in Appendix A. Standard genetic manipulation methods for *S. pombe* transformation and tetrad dissection were used to generate the strains listed. For microscopy experiments, cells were first pre-cultured in 3 mL of Edinburgh minimal media (EMM) in dark conditions at 30 °C for 6–8 h. Once exponentially growing, pre-cultures were diluted (Optical Density (O.D.) _600nm_ = 0.02) in 10 mL of EMM and incubated in dark conditions overnight at 30 °C. In order to allow proper aeration of the culture, 50 mL Erlenmeyer flasks were used. For cell size analyses cells were pre-cultured and diluted once in 3 mL of Edinburgh minimal media (EMM) in dark conditions at 30 °C for 6–8 h. Once exponentially growing, pre-cultures were diluted in 10 mL of EMM and incubated in light conditions for a minimum of 14 h. All live-cell imaging was performed on EMM-ALU agarose pads, except calcofluor-white experiments in which cells were placed directly on a slide [34]. Gene tagging was performed at endogenous genomic locus at the 3′ end, yielding C-terminally tagged proteins, as described [35]. Pak1 gene tagging was performed by transforming a WT strain with AfeI linearized (pBSII(KS^+^))-based single integration vector (pAV72-3′UTR*^pak1^*-Pak1-sfGFP-kanMX-5′UTR*^pak1^*) targeting the endogenous locus [36]. The functional mCherry-tagged and sfGFP-tagged *cdc42* alleles *cdc42-mCherry^sw^* and *cdc42-sfGFP^sw^* were used as described in [4]. Gene tagging, deletion, and plasmid integration were confirmed by diagnostic PCR for both sides of the gene.

The construction of plasmids and strains expressing CIBN-mTagBFP2-Ritc, CRY2, CRY2-Cdc42^∆CaaX^ and CRY2-Cdc42^Q61L,∆CaaX^ was done as described in [6].

To generate the *Pact1-cdc42-mCh^SW^* strain a pINT-ura4^+^ integrative vector was generated. *Pact1* was amplified from gDNA using primers osm2378 (atgggcccgctagcatgcGATCTACGATAATGAGACGG) and osm2379 (ccggctcgagGGTCTTGTCTTTTGAGGGTT) and cloned using ApaI and XhoI. *Cdc42-mCh^SW^-nmt_terminator_* was amplified from pSM1224 using osm2343 (cccaagcttATGCCCACCATTAAGTGTGTCG) and osm2344 (gctctagaCTTCTAATTACACAAATTCCG) and cloned using HindIII and XbaI. As a results pSM1449 was generated. This plasmid was linearized with AvrII and integrated at *ura4* locus of YSM485 strain. The endogenous allele of *cdc42* was deleted using a hygromycin (*hph^+^*) resistance cassette as described [35]. *Hph^+^* cassette was amplified from pSM693 using osm2511 (TACTTAGGGGTTTGAACTTTCTAGGAATTCAATAAAGTGAAGCAAAGCTTTACGATTAATTATTTTTTGTGAAATAGTcggatccccgggttaattaa) and osm2512 (AAGCTAAGACATTGTTTACTGTTGTAAACTAGCTGTATTAAGGAAATTTCGGAAAAGGAAAGAAAACCAGGGGTTAAAgaattcgagctcgtttaaac). Finally, strain YSM3732 was generated by transforming the *Cdc42-mCh^SW^-nmt_terminator_* strain with the *hph^+^* resistance cassette.

In primer sequences, restriction sites are underlined. Plasmid maps are available upon request.

### 3.2. Cell Length and Width Measurements

For cell length and width measurements shown in Figure 4 and Appendix A, cells were grown at 30 °C in 10 mL EMM in light and dark conditions respectively. In this case, we used white light, which we found is sufficient to activate CRY2, shown by the cell rounding triggered by CRY2-Cdc42^Q61L,∆CaaX^. Exponentially growing cells were stained with calcofluor to visualize the cell wall and imaged on a Leica epifluorescence microscope with 60× magnification platform described previously [34]. Measurements were made with ImageJ on septating cells. For each experiment, strains with identical auxotrophies were used.

Aspect ratios were calculated as:(1)Aspect ratio=Cell lengthCell width

The index of ratios shown in Figure 4B was calculated as:IndexCdc42∆CaaX=Aspect ratio(CRY2Cdc42∆CaaX)Aspect ratio (CRY2)
(2)IndexCdc42Q61L∆CaaX=Aspect ratio(CRY2Cdc42∆Q61LCaaX)Aspect ratio (CRY2)

### 3.3. Microscopy

Fluorescence microscopy experiments were done in a spinning disk confocal microscope, essentially as described [34,37]. Image acquisition was performed on a Leica DMI6000SD inverted microscope equipped with an HCX PL APO 100X/1.46 numerical aperture oil objective and a PerkinElmer Confocal system. This system uses a Yokagawa CSU22 real-time confocal scanning head, solid-state laser lines and a cooled 14-bit frame transfer EMCCD C9100-50 camera (Hamamatsu) and is run by Volocity (PerkinElmer). When imaging strains expressing the CRY2-Cdc42^∆CaaX^ and/or CRY2 systems, an additional long-pass color filter (550 nm, Thorlabs Inc., Newton, NJ, USA) was used for bright-field (BF) image acquisition, in order to avoid photo-activation by white light.

Spinning disk confocal microscopy experiments shown in Figure 1, Figure 2 and Figure 3, Appendix A were carried out using cell mixtures [6]. Cell mixtures were composed by one strain of interest (the sample optogenetic strain, expressing or not an additional GFP-tagged protein) and 2 control strains, namely:(1)RFP control: An RFP bleaching correction strain, expressing cytosolic CRY2PHR-mCherry.(2)GFP control: A wild type strain expressing the same GFP-tagged protein as the strain of interest but without the optogenetic system. This strain was used both as negative control for cell side re-localization experiments and as GFP bleaching correction strain (in Figure 1, Figure 2, Figure 3, Figure 5 and related Appendix A).

Strains were handled in dark conditions throughout. Red LED light was used in the room in order to manipulate strains and to prepare the agarose pads. Strains were cultured separately. Exponentially growing cells (O.D._600nm_ = 0.4–0.6) were mixed with 2:1:1 (strain of interest, RFP control and GFP control) ratio, and harvested by soft centrifugation (2 min at 1600 rpm). The cell mixture slurry (1 µl) was placed on a 2% EMM-ALU agarose pad, covered with a #1.5-thick coverslip, and sealed with VALAP (vaseline, lanolin, and paraffin). Samples were imaged after 5–10 min of rest in dark conditions.

The plasma membrane recruitment dynamics of CRY2-Cdc42^∆CaaX^ and CRY2 systems were assessed using cell mixtures. Protein recruitment dynamics were assessed by applying the 3 different photo-activating cycles listed below. Lasers were set to 100%. Shutters were set to maximum speed and in all instances the RFP channel was imaged first, before the GFP channel. The duration of the experiment was equal regardless of the exposure time settings (≈ 15 s):(1)50 ms: RFP channel (200 ms), GFP channel (50 ms). This constitutes one cycle (≈ 0.5 s). 30 time points were acquired (≈ 0.5 s * 30 = 15.1 s).(2)250 ms: RFP channel (200 ms), GFP channel (250 ms). This constitutes one cycle (≈ 0.7 s). 22 time points were acquired (≈ 0.7 s * 22 = 15.1 s).(3)500 ms: RFP channel (200 ms), GFP channel (500 ms). This constitutes one cycle (≈ 0.9 s). 17 time points were acquired (0.9 s * 17 = 15.5 s).

Endogenous GFP-tagged protein re-localization experiments were carried out using cell mixtures. Lasers were set to 100%; shutters were set to sample protection and in all instances the RFP channel was imaged first and then the GFP channel. RFP exposure time was always set to 200 ms, whereas the GFP exposure time varied depending on the monitored protein. Cells were monitored in these conditions for 87 s.

Spinning disk confocal time (sum) projections of five consecutive single plane images are shown in Appendix A. Z-stack images were acquired on a Spinning disk confocal microscope using an optimal z-spacing of 0.71 µm between successive stacks. 6 stacks were acquired (covered Z distance = 4 µm). In Figure 5, single plane GFP and RFP images were combined to generate the merged (Figure 5B) and used for the GFP signal analyses at the cell side (Figure 5C). Cells were cultured as stated above and placed on EMM-ALU pads. Pads were kept for 30 min in the dark or under white-light illumination for 30 min prior imaging.

### 3.4. Image Analysis

All image-processing analyses were performed with Image J software (http://rsb.info.nih.gov/ij/). Image and time-lapse recordings were imported to the software using the Bio-Formats plugin (http://loci.wisc.edu/software/bio-formats). Time-lapse recordings were aligned using the StackReg plugin (http://bigwww.epfl.ch/thevenaz/stackreg/) according to the rigid body method. All optogenetic data analyses were performed using MATLAB (R2019a), with scripts developed in-house. Figures were assembled with Adobe Photoshop CC2019 and Adobe Illustrator 2020.

#### 3.4.1. CRY2-Cdc42^∆CaaX^ and CRY2 Quantifications

The plasma membrane recruitment dynamics of CRY2-Cdc42^∆CaaX^ and CRY2 systems was assessed by recording the fluorescence intensity over a 15-pixel long by 36-pixel wide ROI (roughly 1.25 µm by 3 µm), drawn perpendicular to the plasma membrane of sample cells, from outside of the cell towards the cytosol. The fluorescence intensity values across the length of the ROI were recorded over time in the RFP channel, in which each pixel represents the average of the width (36 pixels) of the ROI (3 replicates, 30 cells per replicate). Average background signal was measured from tag-free wild-type cells incorporated into the cell mixture. The total fluorescence of the control RFP strain was also measured over time in order to correct for mCherry fluorophore bleaching. In both cases, the ROI encompassed whole cells, where ROI boundaries coincide with the plasma membrane.

Photobleaching correction coefficient was calculated by the following formula:(3)RFP bleaching correction coefficient=(RFP Intensitytn− NoGFPBckgtn)(RFP Intensityt0−NoGFPBckgt0)
where *RFP intensity* is the signal measured from single RFP control cells, *NoGFPBckg* is the average background signal measured from tag-free cells, t_n_ represents a given time point along the time course of the experiment and t_0_ represents the initial time point (*n* = 30 time points). These coefficients were corrected by a moving average smoothing method (moving averaged values = 5). RFP bleaching correction coefficient values calculated for individual RFP control cells were averaged in order to correct for bleaching of the RFP signal.

The fluorescence intensity values of optogenetic cells were corrected at each time point with the following formula:(4)$$RFP\;intensity=((Raw\;RFP\;signal_{tn}-\;NoGFPBckg_{tn})/RFP\;bleaching\;correction\;coefficient{}_{tn})$$
where *Raw RFP signal* is for the RFP values measured from sample strains, *NoGFPBckg* is the average background signal measured from tag-free cells and t_n_ represents a given time point along the time course of the experiment (*n* = 30 time points). The profiles resulting from these analyses were used to get the net plasma membrane recruitment profiles (Appendix A, [6]), the fluorescence intensities from the peak ± 1 pixel were averaged and plotted over time.
(5)Peak RFP intensitytn= (RFP intensitypeak−1pixel tn+RFP intensitypeak tn+RFP intensitypeak+1pixel tn)3
(6)$$Net\;P.M.\;recruitment\;Profile=\;(Peak\;RFP\;intensity_{tn}-\;Peak\;RFP\;intensity_{t0})$$

Finally, the single-cell plasma membrane recruitment half-times were calculated by fitting the normalized recruitment profiles with the following formula:(7)$$RFP\;intensity\;\left(y\right)=a*\left(1-e^{\left(-b*t\right)}\right)$$
(8)Recruitment t1/2= ln(0.5)b

#### 3.4.2. Quantifications of the Re-Localization of GFP-Tagged Proteins to Cell Sides

Endogenous GFP-tagged protein re-localization was assessed upon photo-activation of CRY2-Cdc42^∆CaaX^ and CRY2 systems by recording the fluorescence intensity over a 3 pixel-wide by 36 pixel-long (≈ 0.25 µm by 3 µm) ROI drawn parallel to the cell side cortex of sample cells. The average fluorescence intensity values of both GFP and RFP channels were recorded over time from sample strains. In these particular experiments, a GFP control strain was included. These strains serve 2 purposes: (1)Calculation of the GFP bleaching correction coefficient (see below).(2)Negative control of the experiment: These strains carry the same endogenous GFP-tagged protein as the sample strain of the experiment, however lacking the optogenetic system. This controlled that GFP fluorescence changes were due to the optogenetic system and not caused by imaging per se. Control GFP strains were imaged in the same pad and analysed in the same way as optogenetic cells.

To derive photobleaching correction coefficients, the average camera background signals (*Bckg*) from 5 cell-free regions was measured as above, and fluorophore bleaching from RFP control and GFP control strains were measured at the cell side of control RFP and control GFP strains, for RFP and GFP channels, respectively.
(9)RFP bleaching correction coefficient=(RFP Intensitytn− Bckgtn)(RFP Intensityt0−Bckgt0)
(10)GFP bleaching correction coefficient=(GFP Intensitytn− Bckgtn)(GFP Intensityt0−Bckgt0)
where *RFP intensity* and *GFP intensity* stand for the signal measured from RFP control and GFP control cells, respectively, t_n_ represents a given time point along the time course of the experiment and t_0_ represents the initial time point (*n* = 30 time points). These coefficients were corrected by a moving average smoothing method, as above.

The fluorescence intensity values of optogenetic cells in both GFP and RFP channels were independently analysed as follows. First, GFP and RFP signals were background and bleaching corrected, using Formulas (9) and (10) for the RFP and GFP channels, respectively:(11)$$P.M.\;GFP/RFP\;value_{tn}=((Raw\;signal_{tn}-\;Bckg_{tn})/bleaching\;correction\;coefficient{\;}_{tn})$$
where *Raw signal* intensity represents the GFP or RFP raw values at the cell side cortex, *Bckg* stands for the average fluorescence intensity of 5 independent cell-free regions and t_n_ represents a given time point along the time course of the experiment (*n* = 30 time points). The net fluorescence intensity at the cell side cortex was then calculated for both GFP and RFP signals.
(12)$$Net\;P.M.\;GFP/RFP\;value_{tn}=\;(Fluorescence\;intensity_{tn}-\;Fluorescence\;intensity_{t0})$$

From here on, RFP and GFP signals were treated differently. Single cell plasma membrane RFP profiles from Equation (10) were individually normalized and fitted to the Equation (5) in order to extrapolate the parameter b. Using the Equation (6), recruitment half times of CRY2 and CRY2-Cdc42^∆CaaX^ were calculated. Because of lower signal-to-noise of the weak GFP fluorescence, plasma membrane GFP profiles from Equation (10) were averaged (*n* > 20 profiles per experiment). Measurements from cells obtained in 3 experimental replicates are plotted on Figure 1C (CRY2-Cdc42^∆CaaX^
*n* = 84 cells; CRY2 control *n* = 84 cells, WT control *n* = 252 cells); Figure 1E (CRY2-Cdc42^∆CaaX^
*n* = 89 cells, CRY2 control *n* = 78 cells, WT control *n* = 247 cells); Figure 1G (CRY2-Cdc42^∆CaaX^
*n* = 90 cells, CRY2 control *n* = 85 cells, WT control *n* = 280 cells); Figure 2B (Scd2-GFP *scd1∆*: CRY2-Cdc42^∆CaaX^
*n* = 67 cells, CRY2 control *n* = 71 cells, WT control *n* = 226 cells; CRIB-3GFP *scd1∆*: CRY2-Cdc42^∆CaaX^
*n* = 75 cells, CRY2 control *n* = 69 cells, WT control *n* = 222 cells; Pak1-sfGFP *scd1∆*: CRY2-Cdc42^∆CaaX^
*n* = 78 cells, CRY2 control *n* = 67 cells, WT control *n* = 231 cells), Figure 2E (CRY2-Cdc42^∆CaaX^
*n* = 74 cells, CRY2 control *n* = 86 cells, WT control *n* = 247 cells); Figure 3B (CRY2-Cdc42^∆CaaX^
*n* = 79 cells, CRY2 control *n* = 72 cells, WT control *n* = 228 cells); Figure 3E (CRY2-Cdc42^∆CaaX^
*n* = 81 cells, CRY2 control *n* = 75 cells, WT control *n* = 257 cells); Figure 3G (CRY2-Cdc42^∆CaaX^
*n* = 86 cells, CRY2 control *n* = 85 cells, WT control *n* = 246 cells) and Figure 5A (CRIB-3GFP *rga4∆rga6∆*: CRY2-Cdc42^∆CaaX^
*n* = 83 cells, CRY2 control *n* = 80 cells, WT control *n* = 145 cells; Pak1-sfGFP *rga4∆rga6∆*: CRY2-Cdc42^∆CaaX^
*n* = 83 cells, CRY2 control *n* = 84 cells, WT control *n* = 163 cells).

#### 3.4.3. Quantifications of the Re-Localization of GFP-Tagged Proteins from Cell Tips

Scd2-GFP, CRIB-3GFP, Pak1-sfGFP and Scd1-3GFP tip signal analyses were performed from the same time-lapse recordings as cell side re-localization experiments. GFP tip signals were recorded over a 3 pixel-wide by 6–12 pixel-long (≈ 0.25 µm by 0.5–1 µm) ROI drawn at the tip of the cells. To derive photobleaching correction coefficients, the average camera background signals (*Bckg*) from 5 cell-free regions was measured as before, and GFP bleaching from GFP control strain was measured at the cell tip.
(13)Tip GFP bleaching correction coefficient=(GFP Intensitytn− Bckgtn)(GFP Intensityt0−Bckgt0)
where *GFP intensity* stands for the signal measured from the tip of GFP control cells, t_n_ represents a given time point along the time course of the experiment and t_0_ represents the initial time point (*n* = 30 time points). This coefficient was corrected by a moving average smoothing method, as before.

The tip GFP fluorescence intensity values of optogenetic cells was analysed as follows. First, GFP signals was background and bleaching corrected, using Formula (14):(14)$$Tip\;GFP\;value_{tn}=(Tip\;Raw\;signal_{tn}-\;Bckg_{tn})/Tip\;GFP\;bleaching\;correction\;coefficient{\;}_{tn}$$
where *Tip Raw signal* intensity represents the GFP raw values at the cell tip, *Bckg* stands for the average fluorescence intensity of 5 independent cell-free regions and t_n_ represents a given time point along the time course of the experiment (*n* = 30 time points). The tip fluorescence intensities of single optogenetic strains were then normalized relative to their GFP values at the initial time-point.
(15)$$Normalized\;tip\;GFP\;value_{tn}=\left(Tip\;GFP\;value_{tn}/tip\;GFP_{t0}\right)$$

Measurements from cells obtained in 3 experimental replicates are plotted on Figure 1D (CRY2-Cdc42^∆CaaX^
*n* = 88 cells; CRY2 control *n* = 87 cells, WT control *n* = 146 cells); Figure 1F (CRY2-Cdc42^∆CaaX^
*n* = 79 cells; CRY2 control *n* = 67 cells, WT control *n* = 147 cells): Figure 1H (CRY2-Cdc42^∆CaaX^
*n* = 72 cells; CRY2 control *n* = 68 cells, WT control *n* = 203 cells); Figure 2C (Scd2-GFP *scd1∆*: CRY2-Cdc42^∆CaaX^
*n* = 68 cells, CRY2 control *n* = 63 cells, WT control *n* = 134 cells; CRIB-3GFP *scd1∆*: CRY2-Cdc42^∆CaaX^
*n* = 65 cells, CRY2 control *n* = 67 cells, WT control *n* = 131 cells); Figure 2F (CRY2-Cdc42^∆CaaX^
*n* = 69 cells; CRY2 control *n* = 80 cells, WT control *n* = 160 cells); Figure 3C (CRY2-Cdc42^∆CaaX^
*n* = 109 cells; CRY2 control *n* = 75 cells, WT control *n* = 259 cells) and Figure 3F (CRY2-Cdc42^∆CaaX^
*n* = 85 cells; CRY2 control *n* = 78 cells, WT control *n* = 226 cells).

#### 3.4.4. Quantifications of CRIB-3GFP and Cdc42-mCh^SW^ Relative Expression and Distribution Profiles

CRIB-3GFP fluorescence intensity was measured from sum projection of 6 Z-stacks (Appendix A). The background signal from cell-free regions were used to correct the data. The relative fluorescence intensities were calculated by dividing the single-cell CRIB-3GFP fluorescence intensity measurements of WT and *Pact1-cdc42-mCh^SW^* cells by the average CRIB-3GFP fluorescence intensity of WT cells.

CRIB-3GFP distribution profiles were generated from sum projection images of 5 middle-sections (Appendix A). 3-pixel wide ROIs were drawn from side to side following cell membrane contour. The background signal from cell-free regions were used to correct the data. Whole tip profiles were split in half based on the pixel position of their maximum CRIB-3GFP intensity (approximately in the middle of the profile), generating 60 half tips. To take into account the CRIB-3GFP expression level, the average CRIB-3GFP profiles from WT and *Pact1-cdc42-mCh^SW^* cells were then normalized by dividing each value by the relative CRIB-3GFP fluorescence intensities values shown in Appendix A.

#### 3.4.5. Quantifications of CRIB-3GFP and Pak1-sfGFP at the Cell Sides of WT and *rga4∆rga6∆* Mutants

CRIB-3GFP and Pak1-sfGFP re-localization at the cell sides of WT and *rga4∆rga6∆* mutants were assessed upon 30 min of photo-activation by recording the average GFP fluorescence intensity over a 3 pixel-wide by 36 pixel-long (≈ 0.25 µm by 3 µm) ROI drawn parallel to the cell side cortex of CRY2-Cdc42^∆CaaX^ and WT control cells. Control GFP strains were imaged in the same pad and analysed in the same way as optogenetic cells: CRIB-3GFP and Pak1-sfGFP fluorescence signals from CRY2-Cdc42^∆CaaX^ and WT control cells were corrected for the average camera background signals, derived from 6 cell-free regions. The WT control strain, which carries CRIB-3GFP or Pak1-sfGFP like the sample strain of the experiment, but lack the optogenetic system, served as negative control. This controlled that GFP fluorescence changes were not caused by imaging per se. Measurements from these cells were also used as background signal and the average GFP value was subtracted from the the CRY2-Cdc42^∆CaaX^ measurements for each experimental replicate. Measurements from cells obtained in 3 experimental replicates are plotted on Figure 5C (CRIB-3GFP WT control dark = 146 cell sides, light = 202 cell sides; CRIB-3GFP CRY2-Cdc42^∆CaaX^ dark = 143 cell sides, light = 207 cell sides; CRIB-3GFP *rga4∆rga6∆* control dark = 145 cell sides, light = 209 cell sides; CRIB-3GFP *rga4∆rga6∆* CRY2-Cdc42^∆CaaX^ dark = 145 cell sides, light = 211 cell sides; Pak1-sfGFP WT control dark = 138 cell sides, light = 230 cell sides; Pak1-sfGFP CRY2-Cdc42^∆CaaX^ dark = 147 cell sides, light = 221 cell sides; Pak1-sfGFP *rga4∆rga6∆* control dark = 143 cell sides, light = 208 cell sides; Pak1-sfGFP *rga4∆rga6∆* CRY2-Cdc42^∆CaaX^ dark = 144 cell sides, light = 222 cell sides).

#### 3.4.6. Cell Size Measurements, Aspect Ratios and Index of Ratios

The aspect ratio of mutant cells was calculated by dividing the cell length by the cell width (Figure 4A). In Figure 4B, the index of ratios was calculated by dividing CRY2 control aspect ratios by CRY2-Cdc42^∆CaaX^ and CRY2-Cdc42^Q61L,∆^CaaX.

Figure were assembled with Adobe Photoshop CS5 and Adobe Illustrator CS5. All error bars on bar graphs are standard deviations. For statistical analysis, in Figure 1, Figure 2 and Figure 3, cumulative GFP signal (addition of GFP signal along the 30 time points of the time-lapse) was calculated from single cell traces of CRY2, CRY2-Cdc42^∆CaaX^ and GFP control cells.

#### 3.4.7. Cell Images and Kymographs

Cell images shown in Figure 1A, Figure 2A–D, Figure 3A–D, and Appendix A were obtained from time lapses that were first aligned using the StackReg (http://bigwww.epfl.ch/thevenaz/stackreg/#Explanations) plugin on image J. Aligned time lapses were corrected for the average camera background by subtracting the average signal from 6 cell-free regions measured over the first frame (t0). Finally, time lapses were corrected for photo-bleaching using the Bleach correction plugin (https://www.embl.de/eamnet/html/bleach_correction.html) on image J. GFP maximum-intensity projections of entire time lapses (30 time points) were used to generate GFP max projection images. GFP max projection images were merged with RFP max maximum-intensity projections, in which the the first frame (t0) was omitted in order to segment the plasma membrane. Prior to blue light exposure CRY2-Cdc42^∆CaaX^ remains cytosolic as shown in Appendix A.

Kymographs shown in Figure 1B were generated with the MultipleKymograph (https://www.embl.de/eamnet/html/body_kymograph.html) ImageJ plugin. A single pixel-wide ROI was drawn along the cortex at the cell side (1 pixel = 0.083 µm).

#### 3.4.8. Statistical Analysis

For statistical analysis, single cell cumulative GFP signals of the entire dataset (3 independent experiments combined) were considered, without averaging. For CRIB analysis, we also performed statistical tests on cumulative GFP signal of the second half of the time lapse relative to either the first half or the second half of the CRY2 control, both of which returned statistically significant values (*p*-values = 0.0008 and 0.0025, respectively). By contrast, a comparison of cumulative GFP signal over the first half (first 45s) with the corresponding time lapse in the control was non-significant (*p*-value = 0.45). We also did these comparisons on the Scd1-3GFP signal with similar results (2nd half of CRY2-Cdc42^∆CaaX^ vs. 1st half *p*-value = 0.024; 2nd half of CRY2-Cdc42^∆CaaX^ vs. 2nd half of CRY2 control *p*-value = 0.0035; 1st half of CRY2-Cdc42^∆CaaX^ vs. 1st half of CRY2 control *p*-value = 0.12). We chose to report the tests on the cumulative signal for the entire time lapse, as these are more conservative, and thus less prone to over-interpretation. Data normality was assessed by the Lilliesfors test and significance by pairwise Kruskal–Wallis analysis. *p* values show significance of differences between CRY2-Cdc42^∆CaaX^ and CRY2 cells, unless indicated otherwise. A *T*-test was used in Figure 4 and Figure 5C. On each box, the central red mark indicates the median, while the bottom and the top edges indicate the 25th and 75th percentiles, respectively. The whiskers extend to the most extreme data points not considering outliers, which are plotted individually using the red ‘+’ symbol. All experiments were done at least three independent times. For clarity of the average traces, the plots are shown with standard error of the mean (SEM) in the main figures, which shows the robustness of the average value. The same plots are available in the supplementary figures with standard deviation (SD) to show the biological variability.

## Figures and Tables

**Figure 1 cells-09-02089-f001:**
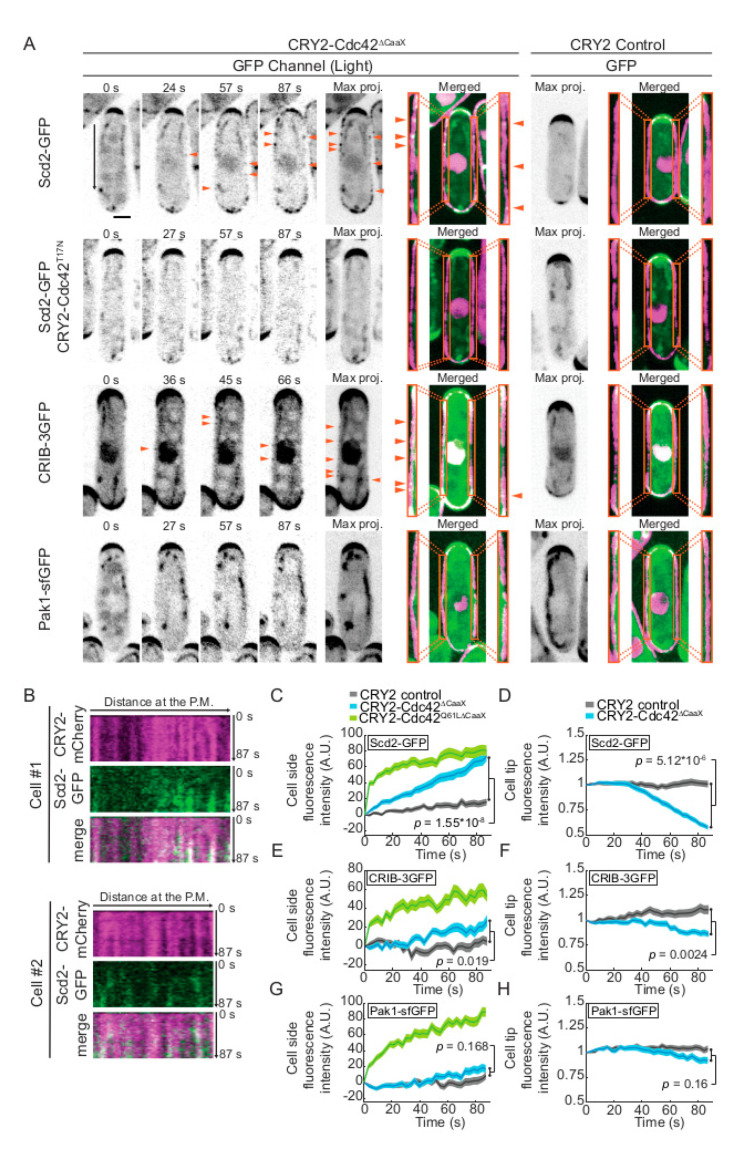
Ectopic sites of Cdc42 activation upon CRY2-Cdc42^∆CaaX^ cell-sides recruitment. (**A**) Localization of Scd2-GFP, CRIB-3GFP and Pak1-sfGFP in CRY2-Cdc42^∆CaaX^-expressing cells (B/W inverted images and green channel in merge). The GFP max projection (“max proj.”) images show GFP maximum-intensity projections of 30 time points over 87 s. Merged images are composites of GFP and RFP max projections (t0 omitted from the RFP projection). Magnification of the lateral cortex is shown in the orange insets. Arrowheads point to lateral Scd2-GFP and CRIB-3GFP signal. The black arrow in Scd2-GFP panel indicates the cortical region in the kymograph shown in (B, cell #1). Note that autofluorescent organelles appear as linear and circular structures in some of the GFP channel images. (**B**) Kymograph at the lateral cell cortex for two cells over the 87 s (cell #1 corresponds to the cell shown in (**A**)). (**C**,**D**) Quantification of Scd2-GFP signal intensity at cell sides (**C**) and cell poles (**D**). (**E**,**F**) Quantification of CRIB-3GFP signal intensity at cell sides (**E**) and cell poles (**F**). (**G**,**H**) Quantification of Pak1-sfGFP signal intensity at cell sides (**G**) and cell poles (**H**). In (**C**–**H**), *n* > 66 cells. Exact numbers are listed in the methods. In (**C**,**E**,**G**), the CRY2-Cdc42^Q61L,∆CaaX^ average trace is shown for cell-side recruitment comparison (data from (Lamas et al., 2020)). In all graphs, thick line = average; shaded area = standard error of the mean (SEM); WT, wild type; A.U., arbitrary units. Bars = 2 µm. Associated trace analysis is shown in Appendix A for cell-side and Appendix A for cell pole analyses.

**Figure 2 cells-09-02089-f002:**
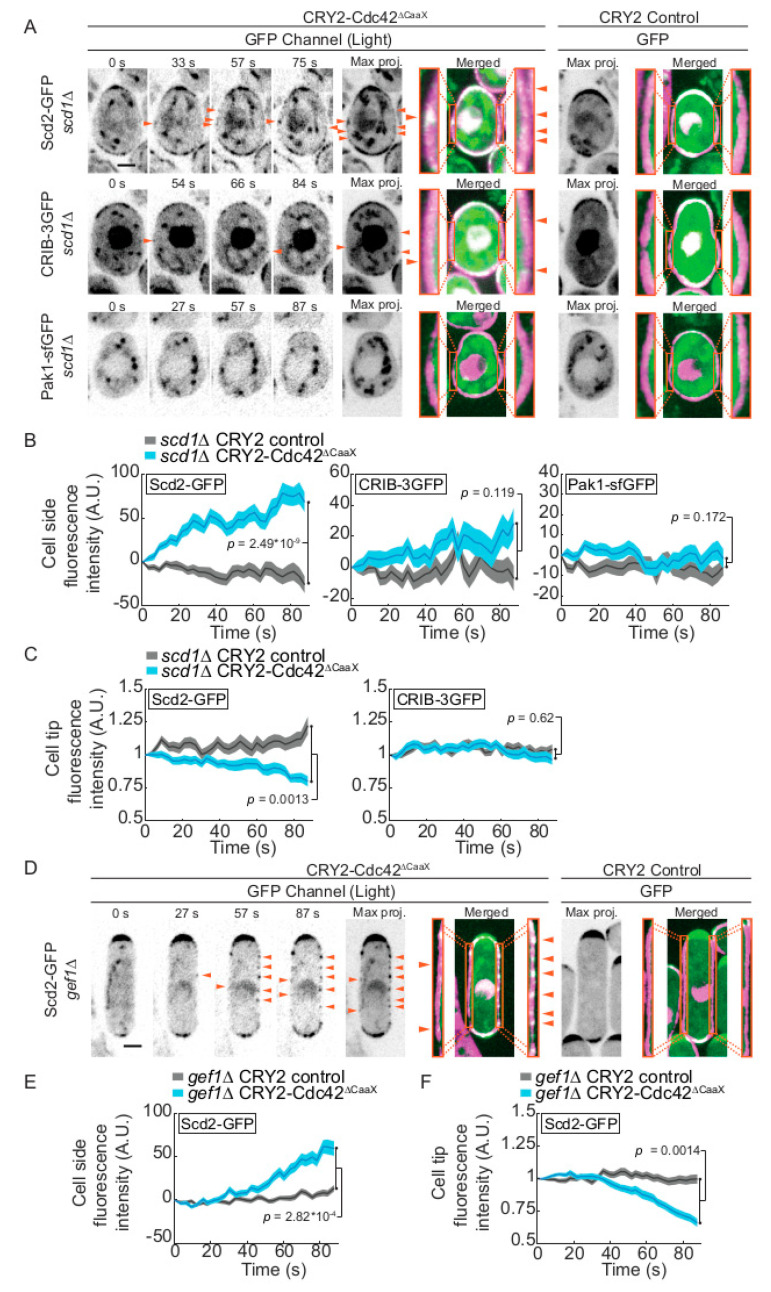
Role of Cdc42 GEFs in activation of CRY2-Cdc42^∆CaaX^. (**A**) Localization of Scd2-GFP, CRIB-3GFP and Pak1-sfGFP in CRY2-Cdc42^∆CaaX^-expressing *scd1∆* cells (B/W inverted images and green channel in merge). The GFP max projection (“max proj.”) images show GFP maximum-intensity projections of 30 time points over 87 s. Merged images are composites of GFP and RFP max projections (t0 omitted from the RFP projection). Magnification of the lateral cortex is shown in the orange insets. Arrowheads point to lateral Scd2-GFP and CRIB-3GFP signal. (**B**) Quantification of GFP signal intensities of Scd2-GFP, CRIB-3GFP and Pak1-sfGFP at cell sides of *scd1∆* mutants. *n* > 66 cells. Exact numbers are listed in the methods. (**C**) Quantification of Scd2-GFP and CRIB-3GFP signal intensity at cell poles of *scd1∆* mutants. *n* > 62 cells. Exact numbers are listed in the methods. (**D**) Localization of Scd2-GFP in CRY2-Cdc42^∆CaaX^-expressing *gef1∆* cells. Layout as in panel A. (**E**,**F**) Quantification of Scd2-GFP signal intensity at cell sides (**E**) and cell poles (**F**) of *gef1∆* mutants. *n* > 68 cells. Exact numbers are listed in the methods. In all graphs, thick line = average; shaded area = standard error of the mean (SEM); WT, wild type; A.U., arbitrary units. Bars = 2 µm. Associated trace analysis is shown in Appendix A. Autofluorescent organelles appear as linear and circular structures in some of the GFP channel images.

**Figure 3 cells-09-02089-f003:**
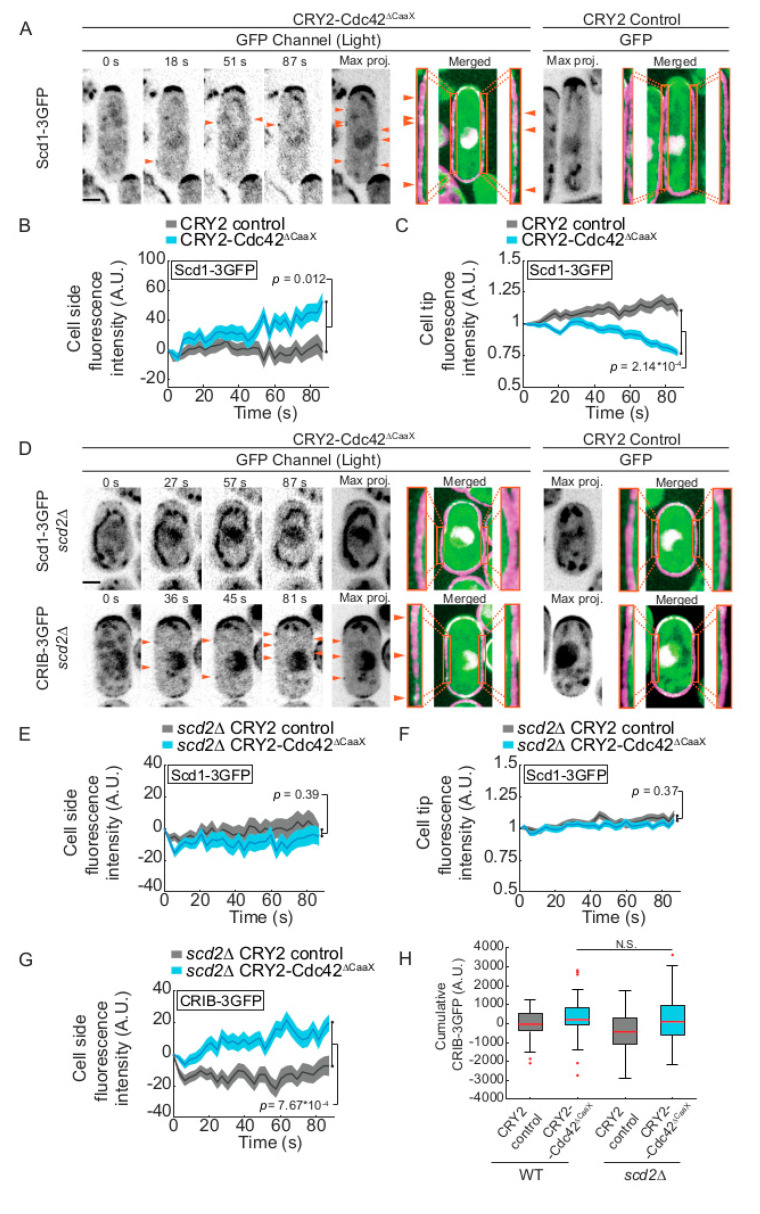
CRY2-Cdc42^∆CaaX^ recruits the GEF Scd1 dependent on the scaffold Scd2. (**A**) Localization of Scd1-3GFP in CRY2-Cdc42^∆CaaX^-expressing cells (B/W inverted images and green channel in merge). The GFP max projection (“max proj.”) images show GFP maximum-intensity projections of 30 time points over 87 s. Merged images are composites of GFP and RFP max projections (t0 omitted from the RFP projection). Magnification of the lateral cortex is shown in the orange insets. Arrowheads point to lateral Scd1-3GFP signal. (**B**,**C**) Quantification of Scd1-3GFP signal intensity at cell sides (**B**) and cell poles (**C**). *n* > 71 cells. Exact numbers are listed in the methods. (**D**) Localization of Scd1-3GFP and CRIB-3GFP in CRY2-Cdc42^∆CaaX^-expressing *scd2∆* cells. Layout as in panel A. (**E**,**F**) Quantification of Scd1-3GFP at cell sides (**E**) and cell poles (**F**) of *scd2∆* mutant. *n* > 74 cells. Exact numbers are listed in the methods. (**G**) Quantification of CRIB-3GFP at cell sides of *scd2∆* mutants. *n* > 84 cells. Exact numbers are listed in the methods. (**H**) Comparison of cumulative CRIB-3GFP intensities in WT and *scd2∆* CRY2 and CRY2-Cdc42^∆CaaX^ cells; in *scd2^+^*
*p*^control^ vs. ^CRY2-Cdc42^ = 0.019 (data from Figure 1E); in *scd2∆*
*p*^control^ vs. ^CRY2-Cdc42^ = 7.67 × 10^−4^ (data from panel 3G); *p*^CRY2-Cdc42 in WT^ vs. *^scd2∆^* = 0.4. In all graphs, thick line = average; shaded area = standard error of the mean (SEM); WT, wild type; A.U., arbitrary units. Bars = 2 µm. Associated trace analysis is shown in Appendix A. Autofluorescent organelles appear as linear and circular structures in some of the GFP channel images.

**Figure 4 cells-09-02089-f004:**
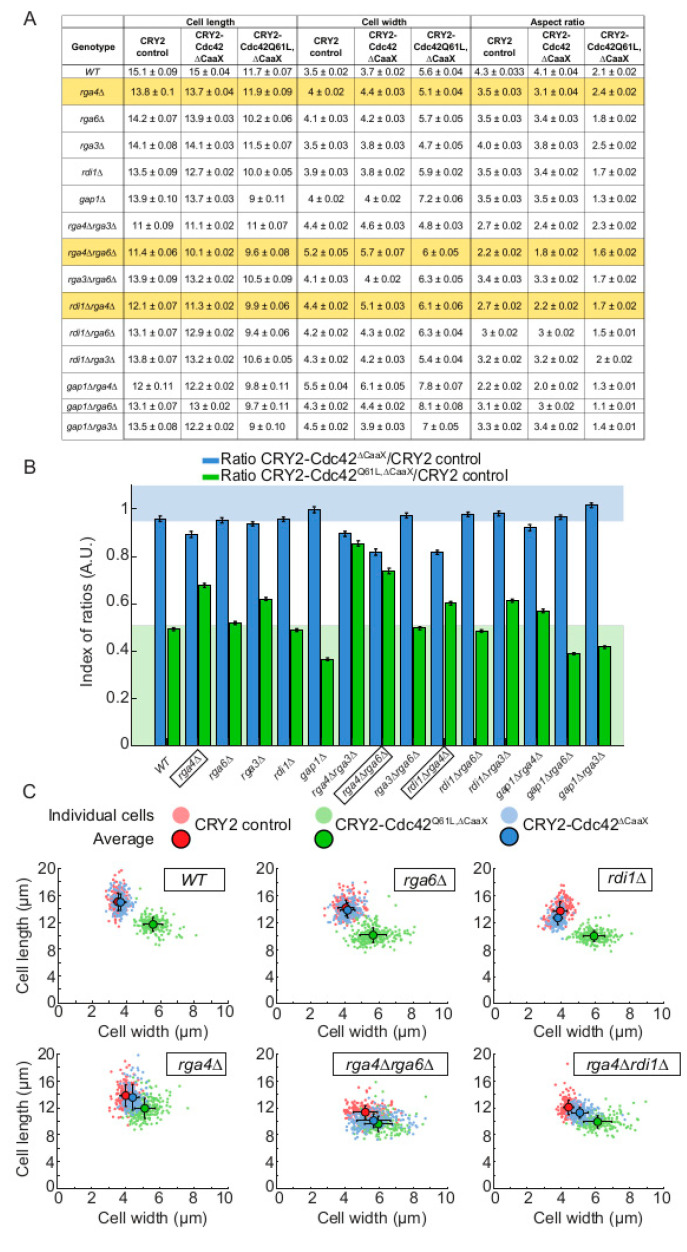
Rga4 GAP prevents growth on cell sides in CRY2-Cdc42^∆CaaX^ cells. (**A**) Average cell length (µm), cell width (µm) and aspect ratio of CRY2, CRY2-Cdc42^∆Caax^ and CRY2-Cdc42^Q61L,∆CaaX^ mutants. For all mutant *N* = 2, *n* > 80 cells per experiment; except for *gap1∆* mutants, *N* = 1, *n* > 80 cells per experiment. (**B**) Aspect ratios of CRY2-Cdc42^∆Caax^ and CRY2-Cdc42^Q61L,∆Caax^ -expressing cells grown in the light normalized to the aspect ratios of CRY2-expressing cells for all the tested mutants. Bars = standard error. The green background indicates expected reduction in aspect ratio upon CRY2-Cdc42^Q61L,∆CaaX^ recruitment in WT cells. The blue background indicates absence of change in aspect ratio upon CRY2-Cdc42^Q61L^ recruitment. Note that all *rga4∆* mutants fall in the white intermediate space. (**C**) Cluster plot of length and width in single cells of WT, *rga4∆, rga6∆, rga4∆rga6∆, rdi1∆ and rga4∆rdi1∆* mutants expressing CRY2, CRY2-Cdc42^∆Caax^ or CRY2-Cdc42^Q61L,∆CaaX^. Small dots = single cells; Large, dark dots = average; bars = standard deviation; A.U., arbitrary units; WT, wild type.

**Figure 5 cells-09-02089-f005:**
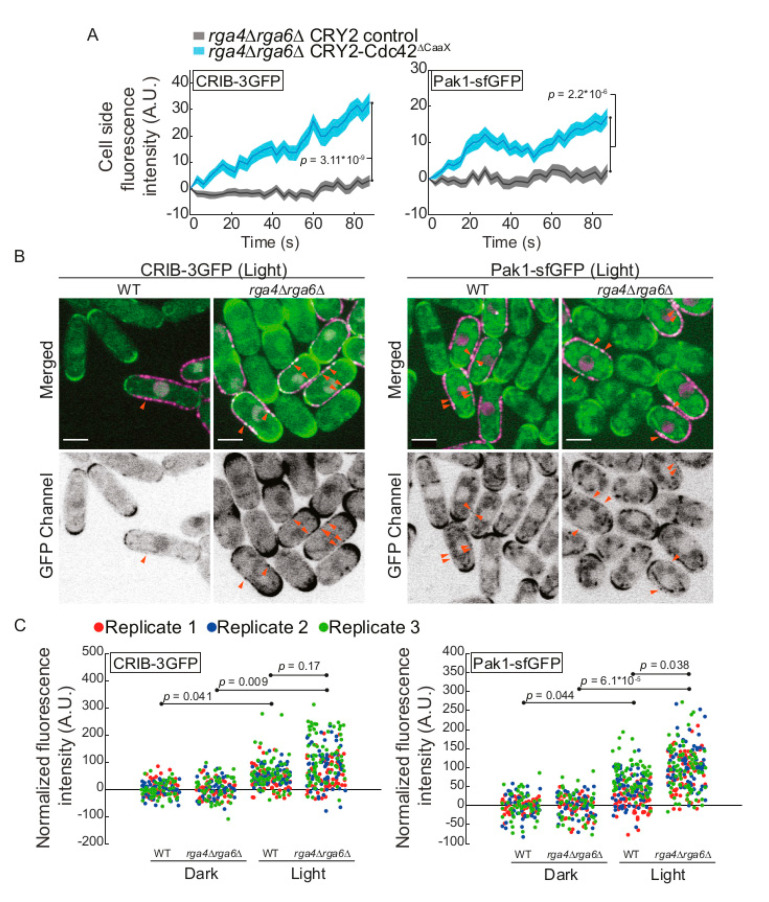
Stronger CRY2-Cdc42^∆CaaX^ activation in cells lacking lateral Cdc42 GAPs. (**A**) Quantification of CRIB-3GFP and Pak1-sfGFP signal intensity at cell sides of *rga4∆rga6∆* mutants, *n* > 80 cells. Exact numbers are listed in the methods. (**B**) Localization of CRIB-3GFP and Pak1-sfGFP in control and CRY2-Cdc42^∆CaaX^-expressing cells upon 30 min-exposure to light (B/W inverted images and green channel in merge). Merged images are composites of GFP and RFP single middle-section images. Arrowheads point to lateral CRIB-3GFP and Pak1-sfGFP signal. Control cells lack the CRY2-Cdc42^∆CaaX^ magenta signal in the merged images. (**C**) Cluster plots of CRIB-3GFP and Pak1-sfGFP signal intensity at cell sides of WT and *rga4∆rga6∆* mutants, *n* > 138 cells. Exact numbers are listed in the methods. The three experimental replicates are plotted with different colours. Statistical *p*-values are from t-test across experimental replicates (*n* = 3). In all graphs, thick line = average; shaded area = standard error of the mean (SEM); WT, wild type; A.U., arbitrary units. Bars = 4 µm. Associated trace analysis is shown in Appendix A. Autofluorescent organelles appear as linear and circular structures in some of the GFP channel images.

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
