# Peer review of "Activation of Cdc42 GTPase upon CRY2-Induced Cortical Recruitment Is Antagonized by GAPs in Fission Yeast"

_cells, 2020, doi:10.3390/cells9092089_

Round 1
Reviewer 1 Report
This article is a sequel manuscript from the previous publication from Martin lab (Lamas et al. 2020). This paper used the same optogenetic tool (CRY2-CIB1) to recruit cytosolic CRY2-Cdc42∆Caax proteins to plasma membrane of fission yeast cells. The authors have adopted identical experimental strategy and careful fluorescence quantifications just like they did in their previous paper. Unlike the previous paper that focused on recruitment of GTP-bound Cdc42Q61L, the authors focused their analysis on wild-type Cdc42. Here the authors made 4 major observations/conclusions:
- Weak activation of CRY2-Cdc42∆Caax upon recruitment to cell cortex
- CRY2-Cdc42∆Caax activation still occurs in absence of either Cdc42 GEF
- CRY2-Cdc42∆Caax promotes recruitment of its GEF Scd1 in Scd2 scaffold-dependent manner
- The Cdc42 GAP Rga4 prevents isotropic growth of CRY2-Cdc42∆Caax cells
However, I am not entirely convinced by the data (at least by the way it is currently presented) in several result sections. In the first result section, the authors claimed that CRIB-GFP Cdc42-GTP reporter is weakly recruited to the cell sides. I have strong reservation on this conclusion, based on the current evidence (see major comment 1). In second result section, the authors concluded that CRY2-Cdc42∆Caax activation on cell sides still occurs in the absence of either Cdc42 GEF, i.e. Scd1 or Gef1. I am somewhat confused by the conclusion drawn (see major comment 4). Contrary to the authors’ conclusion, I think the data presented by the author suggested that CRY2-Cdc42∆Caax activation on cell sides depends on Scd1. This is because CRIB-GFP is no longer recruited to the sides when CRY2-Cdc42∆Caax is recruited to the membrane of scd1∆ cells. The recruitment of Scd2 scaffold protein to the cell sides of scd1∆ cells per se does not constitute a strong evidence of CRY2-Cdc42∆Caax activation upon recruitment.
Throughout the paper, it is clear that the cells have all been subjected to extremely strong laser exposure, because the level of intracellular autofluorescence was immense, especially evident at the first time point. Yet, the signals that the authors detected on the cell sides were so weak and are almost undetectable by naked eyes in most cases. The heavy reliance of the authors on the marginal statistical differences in some experiments in their interpretations, together with the images shown in this paper, cast a strong doubt on whether these kind of extremely weak signals (relative to the tip) can elicit any biologically-relevant, physiological response.
In this paper, the authors focused their analysis on proving that the recruitment of cytosolic CRY2-Cdc42∆Caax “actively” drives growth on cell sides. They claimed that this activity is robustly counteracted by the Cdc42 GAP (Rga4 and Rga6) localized to the cell sides in WT cells. Hence, in the absence of Rga4, CRY2-Cdc42∆Caax recruitment to cell sides can change the morphologies of rga4∆/rga6∆ cells.
However, the authors did not consider an alternative hypothesis that could have explained these observations. That is, when exogenously-expressed CRY2-Cdc42∆Caax is recruited to plasma membrane, they could potentially disrupt the endogenous, active Cdc42-GTP clusters on the cell poles, by sequestering essential scaffold such as Scd2. If this is true, the endogenous Cdc42-GTP cluster must be reduced, but retained significant activities, despite the perturbation by exogenous CRY2-Cdc42∆Caax. This is because WT cells are not morphologically affected by CRY2-Cdc42∆Caax recruitment. In the absence of Rga4 and/or Rga6 GAPs, the reduction in endogenous activity in the Cdc42-GTP clusters on the pole, upon CRY2-Cdc42∆Caax recruitment, can thus additively result major morphological changes in rga4∆ and/or rga6∆ cells. This “passive” mode of perturbation by CRY2-Cdc42∆Caax, rather than the “active” mode of CRY2-Cdc42∆Caax driving growth on cell sides (preferred by the authors), is consistent with a crucial observation in this manuscript, i.e. Pak1, the key effector protein of Cdc42, is never recruited/detected in the cell sides, where CRY2-Cdc42∆Caax is deemed to be active, according to the authors. The authors can potentially refute or lend support to this hypothesis, if careful quantification of the tip signals of Scd1, CRIB and Pak1 are conducted by the authors during revision (see major comment 3).
Major Comments:
- In the CRIB-GFP panel in Figure 1A, the orange arrows are pointing at virtually non-existent foci (apart from the top left orange arrowhead in the 60s time point). Coincidentally, the quantification of the CRIB-GFP signals on the cell side in CRY2 control and CRY2-Cdc42∆Caax suggested that the tiny difference between them is statistically significant. I am not convinced by these data. The difference between the average value is so small (i.e. ~20 a.u.), relative to the standard deviation (i.e.~50-90 a.u.). The p-value of 0.019 can either be on the cusp of statistical “significance” or “insignificance”, depending on where you draw the line. Based on the images shown, one can easily conclude that the difference observed is due merely to noise. I have several recommendations which I hope will strengthen the evidence:
- The authors can choose better images and/or include more timepoints to convince the readers. I’d also encourage the authors to show more cells in supplementary figure to make this statement more convincing. Importantly, the author really needs to show some apparent CRIB-GFP foci, if they are indeed representative of the response of the total population.
- The author mentioned that the CRIB-GFP foci is dynamic (line 104). This is not apparent in this figure preparation. Will a short movie demonstrate this point better?
- The statistical test is done on cumulative GFP signal over 30 timepoints, similar to the previous publication Lamas et al., 2020. In this instance, I think the test over cumulative signal can be misleading for readers not paying attention to the details. For example, in CRIB-GFP, the first 20s, the average signal in CRY2 control and CRY2-Cdc42 overlaps significantly. Are they significantly different? I think the author can consider the possibility of performing statistical tests on two different windows of cumulative intensity, i.e. the first 20s and the last 20s. Hopefully this will show that the statistical test is robust enough to show that there’s no significant difference for the first 20s, and more significant difference can be observed during the last 20s of the imaging.
- The authors should probably increase the sample size in fluorescence analysis, especially in those contentious areas with marginal statistical significance. It is currently unclear how many cells have been analysed in each experiment. Clear annotation will be helpful.
- Between line 107-109, the author described that “The Pak1-sfGFP trace on cell sides also showed an upward trajectory, but was not statistically different from negative control after suggesting delayed recruitment below the detection threshold”. The statistical test on the quantification suggested that the amount of pak1-sfGFP detected on the cell side in CRY2-Cdc42∆Caax is not different from the CRY2 control. Three possible explanations would be: 1. No pak1 recruitment, 2. Minimal pak1 recruitment (below detection threshold) or 3. Delayed pak1 recruitment. Therefore, it is inaccurate to say that the results, by default, imply delayed recruitment of Pak1, because the author did not show that Pak1 is eventually recruited to the cell sides.
- Why are the changes of Scd2, CRIB and Pak1 on cell poles not quantified in the same way you did for cell sides? These data are very important to support/refute a hypothesis that the authors have not considered in this manuscript. It is plausible that by recruiting exogenous CRY2-Cdc42∆Caax to cell sides, the endogenous Cdc42-GTP clusters on the cell poles are disrupted (e.g. due to sequestrations of essential scaffold, effectors etc). This weaker endogenous Cdc42-GTP may not cause gross morphological changes in WT cell, but can present some polarity defect additively in rga4∆ and/or rga6∆
- I disagree with the conclusion drawn in the result section “CRY2-Cdc42 activation still occurs in absence of either Cdc42 GEF”. Line 146-147 state that “Scd2-GFP was still recruited to cell sides (Fig 2A-B), indicating Scd1 is not the sole activator of Cdc42 in these cells”. Paragraphs and its respective quantification preceding this sentence led me to understand that, in scd1∆ cells, CRY2-Cdc42∆Caax recruitment resulted in Scd2 localization to the cell sides, but not CRIB-GFP. Therefore, the lack of CRIB localization to cell sides indicate that CRY2-Cdc42∆Caax is NOT “activated” on the sides of scd1∆ Put simply, Scd1 is crucial or the main activator of side Cdc42 activities. Because of this, I am confused as to why the author inferred that the recruitment of Scd2 indicates that Scd1 is NOT the sole activator of Cdc42 in these cells. Do the authors think that Scd2 recruitment to the side indicates Cdc42 activation, despite the lack of CRIB reporter recruitment? If yes, what is the justification for this inference?
- In Figure 3A, I cannot see any signals in where the orange arrowheads are pointing at. The quantification in Fig.3B shows that there is a statistically-significant difference (p-value= 0.012) in the cumulative GFP signals between CRY2-Cdc42∆Caax and the control. Again, this difference is so marginal, and together with the images shown, I cannot possibly conclude with strong confidence that there is Scd1-GFP on cell sides upon CRY2-Cdc42∆Caax
- In all CRY2-Cdc42 experiment described in this manuscript, Pak1 is not recruited to the cell sides. Pak1 is the key effector of Cdc42 GTPase. How do you propose that morphological changes in rga4∆ /rga6∆ can be effected by CRY2-Cdc42∆Caax without recruiting its major effector kinase (with reference to Fig. 4)?
Minor comments:
- “Cluster a cytosolic Cdc42 allele” (line 14). I think the author refers to clustering cytosolic Cdc42 PROTEIN, because allele is a gene. To mention a gene/allele being activated implies some changes in gene expression.
- “…suggest that the heterologous allele of Cdc42 within…” (line 110). Do you mean exogenously-expressed chimeric CRY2-Cdc42∆Caax PROTEIN?
- Majority of the cells in the RFP channel Fig1A, Fig2C and Fig. 3A has no signal on the growing cell tips. Are growing cell tips depleted of RitC-anchor, or defective recruitment of CRY2-Cdc42∆Caax -mCherry to the growing tip? Any comment on this?
- Line 151-152 states that “The lethality of scd1Δ gef1Δ double mutants unfortunately does not allow testing of this hypothesis (Coll et al., 2003; Hirota et al., 2003).” One can definitely think of knocking-down either Scd1 or Gef1 using thiamine-repressible promoters in the knock-out of its counterpart (e.g. nmtGef1 scd1∆ or nmtScd1 gef1∆). That should drive the remaining GEF activity down to a level mimicking scd1∆ gef1∆ double knock-out. Therefore, I think this hypothesis is rather testable, though I fully appreciate that the current pandemic does not allow for such interrogative revision experiments.
- Line 173-174 states that “Indeed, Scd1 formed weak, dynamic foci at cell sides upon blue-light activation, similar to the CRIB-3GFP foci described above (Fig 3A-B)”. Which CRIB-3GFP foci do you refer to here? The CRIB in Fig.1(barely visible, but statistically-significant) or Fig.2(weak CRIB foci, but not statistically-significant)? Please specify.
- Figure 4A is a very complex table. So, it should not be presented as a figure. I would recommend it to be presented as a graph with box-whisker plot (raw data points) to make it easier for readers to compare the values, or put some of the key numbers in Fig4C to simplify the figure.
- Line 226-227 states that “We then measured the cell length and cell width of calcofluor-stained dividing cells after at least 14h of exponential growth in light conditions and calculated aspect ratios (Fig 4A).” Do you use blue light or white light throughout the 14hr growth? If the experiment is done at white light, is it known how efficient is the Cdc42 recruitment to the sides?
- Missing reference in line 257: (Miller et al, BioRxiv)
Reviewer 2 Report
This manuscript uses an elegant optogenetic system to test the effects of recruiting Cdc42 to the sides of fission yeast cells. This system is nice because the sides of fission yeast cells typically lack Cdc42 activation or downstream cell polarity events. The recruitment is nicely controlled by using a Cdc42 allele that lacks the membrane-binding CAAX motif, and therefore membrane association only happens upon light stimulation. The authors perform a series of experiments wherein they recruit Cdc42∆CAAX throughout the plasma membrane including cell sides, and then use microscopy to assay recruitment of CRIB-GFP (a biosensor for activated Cdc42) and downstream components (Scd1, Scd2, and Pak1). The main conclusion from these experiments is that recruitment of Cdc42 to cell sides is not sufficient to trigger full activation of the signaling pathway. It should be noted that a small amount of Cdc42 activation may be present at cell sides in these experiments, which the authors attribute to clustering from the CRY2 optogenetic construct. However, I was not fully convinced on these points, as noted below. Based on these experiments, the authors next test the role of Cdc42 GAPs, which line the cell sides to prevent ectopic Cdc42 activation at these sites. In a nice series of experiments, they show that optogenetic recruitment of Cdc42∆CAAX in these mutants leads to changes in cell width/morphology, unlike in the wild type cells. This result strongly supports a model in the field wherein the GAPs prevent activation of Cdc42 at these ectopic sites along cell sides.
Many aspects of this paper will be of interest to other researchers. The optogenetic system is very powerful and nicely deployed. The use of different Cdc42 constructs allows a nice comparison of the roles of membrane-binding and activation. I am concerned that two of the main conclusions in the paper are not fully supported by the data. In particular, I do not see much activation of Cdc42 in these cells based on the CRIB biosensor, and I do not see an experiment that clearly tests the role of clustering in Cdc42 activation. It would be nice to see additional data covering these points, along with either revised interpretation or further explanation.
Major concerns:
- I am not convinced that Cdc42 is activated when recruited at the cell sides. The supplemental figures nicely show that Cdc42∆CAAX is heavily recruited to the membrane. However, the CRIB-3GFP signal at cell sides is not convincing to me. The very weak puncta that are highlighted by arrows might be more convincing if they could be shown to colocalize with Scd1 or Scd2. My concern is strengthened by the lack of Pak1 (and presumably other effectors) at cell sides in these experiments. Perhaps a different form of microscopy (e.g. TIRF) to focus on the lateral sides might help? As presented, I was left unclear whether any Cdc42 activation was occurring or not.
- I am confused how the authors conclude that clustering of Cdc42 is leading to activation. As noted in the previous point, evidence for activation is not strong. It would also seem that a non-clustering form of CRY2 might be needed to draw this conclusion. Adding to my confusion, a previous paper from this lab showed that Cdc42 fused to a transmembrane domain led to CRIB-GFP signal at cell sides (Figure 5C in Bendezu et al., 2015). This previous result would suggest that clustering may not be a key step for activation at cell sides, and perhaps something conformational in the CRY2-Cdc42∆CAAX interaction drives the potential appearance of Cdc42 activation, as opposed to clustering.
- Several aspects of the experiments on Cdc42 GEFs were not clear to me. In scd1 mutants, Scd2 is recruited to cell sides but CRIB-GFP is not. This result would indicate that Scd1 is required for Cdc42 activation at cell sides but not for Scd2 recruitment. I recognize that the mechanism may be unclear, but it seems premature to conclude that “neither Cdc42 GEF is individually required for initial activation of clustered Cdc42…” Rather, it seems that this optogenetic system creates a module that can recruit Scd2 in the absence of Cdc42 activation. As an extension to this point, it would be reassuring to see CRIB-GFP experiments in the gef1 mutant cells instead of relying on Scd2 localization as a proxy for activated Cdc42.
- It would be nice to see CRIB-GFP localization in the GAP mutant experiments. Based on the phenotypes, it seems likely that there is a strong presence of activation Cdc42 at cell sides.
Minor concerns:
- First paragraph of introduction could use a citation e.g. a review.
- It would be helpful to reword the sentence on line 64 page 2 beginning “Because Bem1 also acts as a scaffold…”
- Page 2 Line 93: should say “…rod-shape…”
Reviewer 3 Report
In this manuscript the authors study the mechanisms and consequences of clustering and activation of Cdc42 that was synthetically recruited to the plasma membrane of fission yeast cells via the optogenetic CRY2 system. They closely follow their previous publication where they introduced the optogenetic approach and focused on the core feedback components required for scaffold-mediated cell polarization. The novel aspects here are the focus on the activity of Cdc42 at the cell sides and on the buffering of the feedback loop through the GAP Rga4.
In general the presented data is of very high quality and many results are quantified and presented in a very convincing and innovative manner. However, I was a bit confused by the high overlap of approaches and conclusions compared to the Plos Biology paper of last year. It often did not become completely clear to me, what had already been shown before and what aspects were actually novel. In this regard I think the authors could simply add a few summarizing sentences that clearly delineate the differences between both studies. Apart from this I have a few comments that could be addressed to improve the data and presentation, but I think the manuscript can be published with only little additional experimental work.
Specific comments:
- The authors repeatedly state that deletion of GDI has little effects on Cdc42 dynamics but I could not find any actual data on this in the cited references. Is the turnover of Cdc42 at the polarized site really not slowed down in the absence of the GDI (as it is in budding yeast)? I find it very hard to imagine how a reaction-diffusion system of polarization for Cdc42 (via scaffold and GEF) can actually work without a rapidly diffusing and well-mixed pool of cytosolic Cdc42. And for this geranylgeranylated protein such a pool should require the GDI for solubilization. It would be nice to see some thoughts on this issue in the discussion or introduction.
- Throughout the figures there are strongly stained linear structures in the cells that are not mentioned – they likely represent microtubules or auto fluorescent mitochondria. I would suggest to add a short comment in the figure legend if this is not of further relevance for the interpretation.
- The clusters at the cell sides are very often so weak that I had some trouble actually seeing the m in the figures – maybe some images could be replaced by better examples
- The graphs showing the increase in intensities are an interesting way of showing the results but actually rather confusing – the colour schemes with solid filling of the range of data points is difficult to follow (lines for average values have different colours than the range) – I would strongly suggest to show average lines and ranges with dotted lines of the respective colours instead. Also, the small changes coupled to the huge spread of data is not terribly convincing, considering the central importance of these measurements.
- One key conclusion that the authors repeatedly make is that clusters of polarity regulators (Scd1, Scd2, CRIB) directly reflect recruitment by Cdc42 or some other factor – they never test the alternative that activation of Cdc42 induces conformational changes in the binding partner already in the cytosol (or indirectly through biochemical reactions) and that binding to the plasma membrane is then mediated largely via lipid interaction. This has been shown for budding yeast by the McCusker lab and cooperative interactions between anionic lipids (PS, PI4P, PI4,5P2) and the GEF, Scaffold and Cdc42 itself could drive many of the the observed clusters. The CRIB probe in particular consists of the Cdc42 binding part and a membrane binding PH domain – so some of the observed effects might not be directly mediated by Cdc42. One possible control would be express the cytosolic Cdc42 variant without Cry2 (or even better fused to a non-reacting variant of Cry2), expose to 488nm light and then observe localization of the different markers.
- The Miller et al (BioRxv) citation is missing in the literature list.
Round 2
Reviewer 1 Report
The authors have made significant improvements toward the original manuscript. However, there are still a few important points that I think the authors need to address.
- In Fig. 2C, the graphs show that Scd2-GFP level on cell tips of scd1∆ CRY2-Cdc42∆CAAX (blue line) decreases over time, and CRIB-3GFP remains unchanged. However, when I look at the scd1∆ cells in Fig. 2A, it is very clear that Scd2-GFP and CRIB-3GFP accumulate at the cell tips very strongly following recruitment of CRY2- Cdc42∆CAAX. The level of increase is way more significant/apparent, compared to the signals on cell side. Why are the two results (Fig. 2C and 2A) internally contradictory? Is this the wrong graph?
- I thank the authors for careful quantification of the Scd2, CRIB and Pak1 signals on the cell poles. Fig.1D and Fig.1F both confirm that upon Cdc42∆CAAX recruitment to the plasma membrane, native Cdc42-GTP clusters on WT cell poles are significantly perturbed. Fig.1D shows that almost 50% of Scd2 is removed from the cell pole when Cdc42∆CAAX is recruited to the membrane. Yet, Fig.1D is not mentioned nor discussed anywhere in the revised manuscript. These two pieces of data (i.e. Fig.1D and 1F) ought to be highlighted to the readers, as it can influence how they interpret the subsequent results on rga4∆
- On Page 6, the authors stated “Alternatively, the cell shape change observed in rga4Δ cells may reflect a weakening of polarity at the native sites at cell poles.”. In my opinion, the authors ought to remind the readers that you have actual, strong data (i.e. Fig. 1D and 1F) to support this claim. The way it is currently written make this statement sounds like a hand-wavy hypothesis, with no proof. In fact, the phenotype in rga4∆ cells may well be caused by combination of perturbed native Cdc42-GTP at the pole, and some weak, ectopic activation of Cdc42∆CAAX on the cell sides. Putting this explicitly in the text will help the readers to understand the caveats of these interpretations.
- Standard error of mean, instead of “standard error or mean”. Throughout the revised manuscript, the authors have changed the graphs to show standard error of mean (SEM), rather than standard deviation (SD) used in Lamas et al. 2020 (Plos Biology). I think it’s advisable for the authors to include some justification for the change from SD to SEM in this new manuscript in methods, considering that this paper used identical technologies and analysis pipeline as the previous paper. Without any justification, this gives the readers the false impression of opting for SEM in this study for its visual appeal rather than statistical merit.
Reviewer 2 Report
The authors have changed the data presentation and have edited the text in response to the reviewer comments. In my opinion, the main concerns shared by both reviewers have not been addressed fully, but the authors have edited the text so that it more closely reflects the data. Here are 3 lingering comments:
1. I remain unconvinced that Cdc42 is activated at cell sides based on the CRIB marker. The enhanced contrast provides somewhat better presentation, but it still seems a stretch to conclude that Cdc42 is activated based on this marker. It seems important that both reviewers were not convinced of this central point. 2. The impact of the work would be enhanced by a demonstration that CRIB is recruited to cell sides in the GAP mutants. I understand that covid has prevented the addition of new experiments to the paper, but it would be nice to see robust recruitment of CRIB to the cell sides in these mutants, to connect with the cell shape phenotype. 3. The title implies that clustering is what drives Cdc42 activation, but I still don’t think that the authors have really tested the role of clustering per se. I would advise changing the title to read "CRY2-induced activation…”
